# Intra-agent speech permits zero-shot task acquisition

**Chen Yan**
DeepMind
London, UK
ywc@deepmind.com

**Federico Carnevale**
DeepMind
London, UK
fedecarnev@deepmind.com

**Petko Georgiev**
DeepMind
London, UK
petkoig@deepmind.com

**Adam Santoro**
DeepMind
London, UK
adamsantoro@deepmind.com

**Aurelia Guy**
OpenAI*
San Francisco, USA
7aureliaguy@gmail.com

**Alistair Muldal**
DeepMind
London, UK
alimuldal@deepmind.com

**Chia-Chun Hung**
Isomorphic Labs*
London, UK
aldenhung@google.com

**Josh Abramson**
DeepMind
London, UK
jabramson@deepmind.com

**Timothy Lillicrap**
DeepMind
London, UK
countzero@deepmind.com

**Gregory Wayne**
DeepMind
London, UK
gregwayne@deepmind.com

## Abstract

Human language learners are exposed to a trickle of informative, context-sensitive language, but a flood of raw sensory data. Through both social language use and *internal* processes of rehearsal and practice, language learners are able to build high-level, semantic representations that explain their perceptions. Here, we take inspiration from such processes of "inner speech" in humans (Vygotsky, 1934) to better understand the role of *intra-agent speech* in embodied behaviour. First, we formally pose intra-agent speech as a semi-supervised problem and develop two algorithms that enable visually grounded captioning with little labeled language data. We then experimentally compute scaling curves over different amounts of labeled data and compare the data efficiency against a supervised learning baseline. Finally, we incorporate intra-agent speech into an embodied, mobile manipulator agent operating in a 3D virtual world, and show that with as few as 150 additional image captions, intra-agent speech endows the agent with the ability to manipulate and answer questions about a new object without any related task-directed experience (zero-shot). Taken together, our experiments suggest that modelling intra-agent speech is effective in enabling embodied agents to learn new tasks efficiently and without direct interaction experience.

## 1 Introduction

Contemporary language models learn from troves of text comprising billions, and sometimes trillions of tokens. This is not the same situation faced by embodied language learners, such as human children,

---

*Work done while at DeepMind.

36th Conference on Neural Information Processing Systems (NeurIPS 2022).

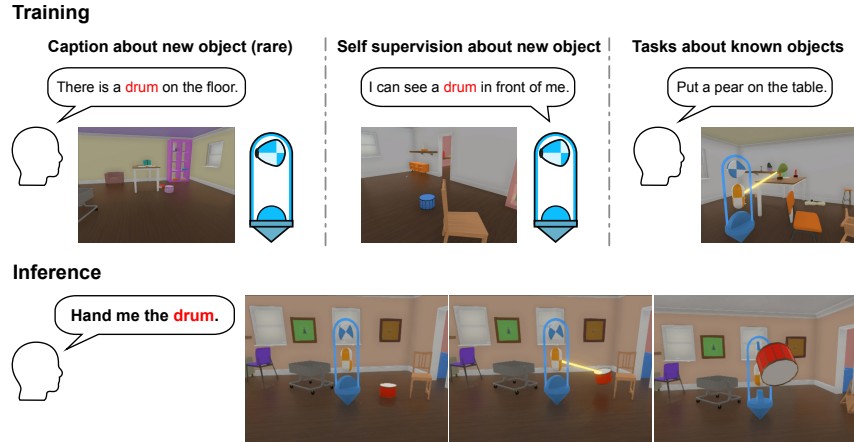

**Figure 1: Learning to speak about new objects permits zero-shot motor task behavior.** Is it possible for agents to perform complex tasks with a new object just by learning to name it? In this work, we show it is. We first trained agents to speak about objects that had never been used in any task, nor named in any interaction, by training agents to caption their observations (top row, left and middle). We then introduced interactive tasks involving other objects and trained agents with imitation learning (top right). Using only a small number of labeled captions that pertained to a new object category (drums), agents were able to perform tasks involving the new objects, such as lifting and delivering them to a second avatar (bottom).

whose raw sensory data vastly exceeds their social-linguistic experiences. Psychologists have long noted the existence of non-social, intra-personal dialogue—such as inner speech (silent) or private speech (aloud)—which provides additional linguistic experiences beyond those encountered in social settings. Indeed, one possible reason for intra-personal speech may be to support the development of social speech [1–5], though the inverse causal relation may also be possible [1, 6].

Intra-personal speech may also affect broader aspects of our cognition: the process wherein we privately speak about our perceptions subsequently impacts how we attend to, remember, or manipulate the objects of our perception [7–12]. For example, our short-term memories tend to commit errors for items that are phonetically but not visually or semantically similar [8, 9], and "self-talk" affects task performance on executive tasks both in children [11, 12] and in adults [7], and has been widely used to improve athletic performance [10].

Inspired by these ideas, here we ask if we can leverage *intra-agent speech* as a tool for artificial intelligence research. Specifically, we ask the following questions: First, is it possible to model intra-agent speech as a machine learning problem, enabling us to make up for a paucity of language data when given access to a large amount of unlabeled data? Second, can mechanisms for learning intra-agent speech impact the subsequent behavior of an embodied agent?

Regarding the first, we formally structure intra-agent speech as a semi-supervised problem. In particular, we model intra-agent speech samples as variational latent variables representing images, and use this perspective to devise two algorithms enabling visual captioning given a paucity of directly labeled data. Regarding the second, we go on to show how embedding such algorithms in an embodied agent not only permits the capacity to speak about any given perception (that is, *to caption*, which is precisely what is being optimized for) but rather also permits zero-shot behaviors involving new objects that were never directly implicated in any task-directed experience (see Figure 1).

## 2 Approach

At a high level, we structured intra-agent speech as the production of language generated from inputs (which in our case manifests as captions of images), with two additional concerns: First, there will be a preponderance of unlabeled data (i.e., pure images) relative to labeled data (i.e., images with associated captions) in our captioning dataset, much like the situations faced by human children in nature and semi-supervised learning in machine learning. Second, the linguistic representations eventually produced will provide auxiliary supervision for an embodied, acting agent so that they may influence interactive behavior.

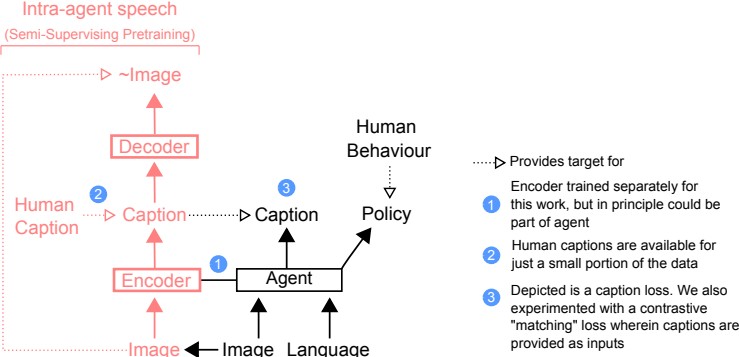

**Figure 2: Using models of intra-agent speech to influence behavior.** During intra-agent speech pre-training, agents receive image observations as input and produce either utterances that describe the observations (for the small number of inputs for which we have human annotated labels) or are tasked with reconstructing the images (using a generative model whose latent representations comprise language tokens). During behavioral training, the intra-agent speech module is frozen and is used to provide targets for an auxiliary captioning loss, whose gradients affect components responsible for action selection.

In humans, the learning of external and internal speech, the rehearsal of inner speech, and the practice of tasks can develop concurrently. In the first stage, we pretrain a captioning model by semi-supervised learning, using a small amount of labeled language data and the large amount of agent's experience as unlabeled data. This model is able to commentate about visual observations – a critical function of "inner speech". This model is then used to commentate (via what we have named "intra-agent speech") on further experiences to shape representations as the agent goes on to learn specific tasks, and we show comparisons of the agent's performance with and without this intra-agent speech process.

A diagram of the semi-supervised learning approach to intra-agent speech and its role in acquiring behaviors is shown in Figure 2.

## 2.1 Methods for Semi-supervised Language Learning

Our data $D$ are composed of a dataset $D_p$ of paired data $\{\mathbf{x}, \mathbf{y}\}$ and a dataset $D_u$ of unpaired data $\{\mathbf{x}\}$. In our interpretation, $\mathbf{y}$ corresponds to language (a caption) describing the image $\mathbf{x}$. When observing unpaired data, we treat $\mathbf{y}$ as a latent variable that must be inferred to explain the observed data [13]. We develop two variants, one generative and one contrastive, of the semi-supervised language learning algorithm in what follows.

### 2.1.1 Generative variant

The preponderance of our data is unlabeled. To model these data we can maximise a variational lower bound on the log marginal distribution, $\log p_\theta(\mathbf{x})$, corresponding to posterior inference. We will use a learned approximation to the true posterior distribution, $q_\omega(\mathbf{y} \mid \mathbf{x})$. We call this approximate distribution an "image-conditional language encoder" (or sometimes, simply, *caption model*) since its functional role is to convert images into a discrete latent code that is, ideally, a semantically meaningful language caption:

$$\sum_{\mathbf{x} \sim D_u} \log p_\theta(\mathbf{x}) \geq \sum_{\mathbf{x} \sim D_u} \mathbb{E}_{q_\omega(\mathbf{y}|\mathbf{x})}[\log p_\theta(\mathbf{x} \mid \mathbf{y})] - \mathcal{D}_{\mathrm{KL}}[q_\omega(\mathbf{y} \mid \mathbf{x}) \| p_\phi(\mathbf{y})] = J_u. \quad (1)$$

In Equation 1, the space of latent codes (captions) is discrete and vast, complicating optimization. However, we can form a stochastic approximation by sampling from $q_\omega(\mathbf{y} \mid \mathbf{x})$ to estimate a gradient.

$$\nabla_\omega J_u = \sum_{\mathbf{x} \sim D_u} \mathbb{E}_{q_\omega(\mathbf{y}|\mathbf{x})}\big[\nabla_\omega \log q_\omega(\mathbf{y} \mid \mathbf{x})\big(\log p_\theta(\mathbf{x} \mid \mathbf{y}) + \log p_\phi(\mathbf{y}) - \log q_\omega(\mathbf{y} \mid \mathbf{x})\big)\big]. \quad (2)$$

For simplicity, we pretrain the prior $p_\phi(\mathbf{y})$ as a language model that learns on all of our caption data, and fix it during this optimisation phase. Thus, optimising $J_u$ corresponds to an entropy-regularised

RL problem [14–16]. The encoder $q_\omega(\mathbf{y} \mid \mathbf{x})$ serves as a policy that achieves reward for matching the language model prior while enabling the language-conditional decoder to reconstruct the image.

While this reward pressures the image-conditional language encoder to produce linguistically meaningful latent codes, we can also exert additional pressure by using a small number of images with associated captions ("labeled data"), and leverage supervised learning to learn $\log p_\theta(\mathbf{x} \mid \mathbf{y}), \log p_\phi(\mathbf{y}), \log q_\omega(\mathbf{y} \mid \mathbf{x})$:

$$J_p = \sum_{\mathbf{x}, \mathbf{y} \sim D_p} \big[ \log p_\theta(\mathbf{x} \mid \mathbf{y}) + \log p_\phi(\mathbf{y}) + \log q_\omega(\mathbf{y} \mid \mathbf{x}) \big]. \tag{3}$$

Over the combined paired and unpaired datasets, our complete objective is to maximise $J = J_p + J_u$.

### 2.1.2 Contrastive variant

The language-conditional image decoder $p_\theta(\mathbf{x} \mid \mathbf{y})$ is a potentially complicated term to model. To understand if this term represents a liability for the approach and to generate an algorithm with potentially complementary strengths, we also developed a contrastive, energy-based approach, amounting to multi-class classification, to approximate it [17–19]. If we have a multi-class classifier that discriminates if a batch element $\mathbf{y}_j$ is paired to a batch element $\mathbf{x}_j$, and we express its cross-entropy loss for the correct index $c = j$ as $L(\{\mathbf{x}_b\}_{b=1}^B, \mathbf{y}_j, c = j)$, then we can show that the generative reconstruction loss $\log p_\theta(\mathbf{x}_j \mid \mathbf{y}_j)$ is proportional to $L(\{\mathbf{x}_b\}_{b=1}^B, \mathbf{y}_j, c = j)$ up to constant factors with respect to $\mathbf{y}$ (Appendix A.1). Thus, we can substitute $L(\{\mathbf{x}_b\}_{b=1}^B, \mathbf{y}_j, c = j)$ for batch element $j$, $\log p_\theta(\mathbf{x}_j \mid \mathbf{y}_j)$, in Equation 2, and the gradients with respect to $\omega$ remain the same in expectation.

We represented the softmax in the classifier as $\frac{e^{\mathbf{f}(\mathbf{x}_j)^\top \mathbf{g}(\mathbf{y}_j)}}{\sum_{b=1}^B e^{\mathbf{f}(\mathbf{x}_b)^\top \mathbf{g}(\mathbf{y}_j)}}$, where $\mathbf{f}$ and $\mathbf{g}$ are networks that compress the image into a vector and the caption into a vector, respectively. To train this classifier, we adopted a similar approach to recent models for visual language representation learning [20, 21] by minimizing the sum of two losses, one for matching each image to its paired caption in the batch, and one for matching each caption to its paired image:

$$L = \frac{1}{B} \sum_{j=1}^B \log \frac{e^{\mathbf{f}(\mathbf{x}_j)^\top \mathbf{g}(\mathbf{y}_j)}}{\sum_{b=1}^B e^{\mathbf{f}(\mathbf{x}_b)^\top \mathbf{g}(\mathbf{y}_j)}} + \frac{1}{B} \sum_{j=1}^B \log \frac{e^{\mathbf{f}(\mathbf{x}_j)^\top \mathbf{g}(\mathbf{y}_j)}}{\sum_{b=1}^B e^{\mathbf{f}(\mathbf{x}_j)^\top \mathbf{g}(\mathbf{y}_b)}} \tag{4}$$

The networks $\mathbf{f}$ and $\mathbf{g}$ are trained at the same time as the rest of the model, and samples from $q_\omega(\mathbf{y}_j \mid \mathbf{x}_j)$ are included as positive examples for the classifier along with the original paired data.

### 2.1.3 Intra-agent speech architecture and optimization

**Image-conditional language encoder.** The image-conditional language encoder received input images with resolution $96 \times 72$. We used the ResNet architecture described in [22], with strides $(1, 1, 2, 2)$, $3 \times 3$ kernel size, and channel sizes $(64, 64, 128, 256)$ for a total of 20 layers. Language was produced by a 4-layer causal transformer with 256 embedding size and 4 attention heads [23], which attended to the 432 hyper-pixel vectors generated from the ResNet, and produced a sequence of logits corresponding to a $4,000$ token vocabulary. Language targets were encoded with a SentencePiece byte-pair encoder [24] trained using language data from [25].

**Generative semi-supervised model.** The language prior comprised a separate transformer pre-trained on all the caption labels in the training data (and hence, did not have image inputs). For the image decoder, we first pre-trained a VQ-VAE [26] on all unlabeled images to define the VQ-VAE's codebook, compressing each image into 432 tokens with a vocabulary of 512. We then used an 8-layer transformer with 512 embedding size with causal masking to model the VQ-VAE tokens autoregressively. The conditioning captions were tokenized and embedded as in the image-conditional language encoder, and then processed by a distinct transformer with 4 layers, 256 embedding size, and 4 heads, whose output was then cross-attended by the image decoder to produce a reconstruction.

**Contrastive semi-supervised model.** For the contrastive model, images were first encoded using a ResNet with the same architecture as the image-conditional language encoder, and globally mean-pooled into one vector with dimension 1024. Language was first encoded using a transformer with

the same 4-layer architecture described above, and then flattened and compressed with a 2-layer MLP with dimensions $(2048, 1024)$.

**Optimization.** We used V-MPO [16] to optimize $q_\omega(\mathbf{y} \mid \mathbf{x})$. The loss was optimized by the Adam optimizer [27] with $\beta_1 = 0.9, \beta_2 = 0.999$ and a learning rate of $2 \times 10^{-4}$ with early stopping at the lowest log-likelihood over captions in validation data. We trained all models with a batch size of $128$ except for the contrastive classifier model, which also received $2,048$ unlabeled images per batch, giving a total batch size of $2,176$. We trained our models using Tensor Processing Units (TPUv3) [28]. In all experiments the early stopping criteria was reached within 150K optimization steps.

## 2.2 Agent training

The agent was trained by behavioral cloning on a large corpus of human interactions in the Playhouse environment [22]. Briefly, it comprises a 3D simulated environment wherein two embodied avatars interact with natural language to cooperatively accomplish tasks involving object manipulation, navigation, and question answering. In addition to receiving visual observations and producing motor actions, agents in this environment also receive language inputs and produce language outputs. This allows them to answer questions, participate in dialogue, ask for clarifications, and so on.

Our approach to training the linguistic representations of the agent therefore has two steps. First, by using semi-supervised language learning, we can amplify the impact of sparse language annotations provided for a small number of images in learning the intra-agent speech module. In turn, using this trained module, we are able to sample captions at high frequency as the agent acts, providing the agent with dense language supervision. Therefore, while the agent trains using behavioral cloning, we can sample captions from the pretrained intra-agent speech module, and provide these as either auxiliary inputs or targets to the agent so that they may influence the agent parameters optimized to produce embodied behaviors.

For our first loss – the caption loss – captions arising as intra-agent speech samples were provided as targets for the agent's language output policy:

$$L_C = -\frac{1}{B} \sum_{b=1}^{B} \sum_{t=0}^{K} \ln \pi_l(\mathbf{y}_{b,t}^c | \mathbf{o}_{b,\leq t}), \tag{5}$$

where $\pi_l$ is the language action policy of the agent, $\mathbf{y}^c$ is the caption sample, $\mathbf{o}$ represents all visual and text observations, and $K$ represents the maximum unroll length. To allow the agent to distinguish whether it was being tasked with emitting a caption of its current observation, or whether it was being tasked with emitting language for the purposes of interactive behavior (see [22]), the input of the language policy is summed with a learnable embedding of the indicator variable representing whether the target is captioning or language output from demonstration.

While the caption loss trains the agent to speak about what it sees, the agent's language encoding (via linguistic inputs) remains untrained. To this end we introduced a second loss: the caption-matching loss. In the caption-matching loss, the agent is required to predict whether a given caption, provided as input, matches the observed image. Negative samples (i.e., captions that do not match the observation) are captions associated with images from elsewhere in the batch [29]. To perform this prediction, an auxiliary classifier $D$ is attached to the network of the agent that encodes both visual and text observations in a separate training pass from the behavioral cloning optimization.

$$L_{CM} = -\frac{1}{B} \sum_{b=1}^{B} \sum_{t=0}^{K} \left[ \ln D(\mathbf{o}_{b,t}^X, \mathbf{y}_{b,t}^c) + \ln \left(1 - D(\mathbf{o}_{b,t}^X, \mathbf{y}_{\text{roll}(b),t}^c)\right) \right]. \tag{6}$$

Here $\mathbf{o}^X$ represents the visual observation, and the caption sample is fed as a text observation. The roll() function rolls the index over batches, so that $y_{\text{roll}(b)}^c$ represents a caption from another batch element as a negative example. The roll function was implemented as $\text{roll}(b) = (b+1) \mod B$.

And finally, the behavioral cloning loss over human language and movement action sequences is as described in Interactive Agents Team [22]:

$$L_{BC} = -\frac{1}{B} \sum_{b=1}^{B} \sum_{t=0}^{K} \left[ \ln \pi_l(\mathbf{a}_{b,t}^l | \mathbf{o}_{b,\leq t}) + \ln \pi_m(\mathbf{a}_{b,t}^m | \mathbf{o}_{b,\leq t}) \right], \tag{7}$$

where the $\pi_m$ and $\mathbf{a}^m$ represents the movement policy and action, respectively.

The total loss is the sum of these losses $L = L_C + L_{CM} + L_{BC}$. Other than the addition of the caption loss and caption-matching loss, we follow Interactive Agents Team [22] for the agent architecture and training hyper-parameters.

## 3 Experiments

### 3.1 Semi-supervised captioning

**Hypothesis.** We first sought to validate our method of semi-supervised learning. We hypothesized that with the additional unlabeled images, the semi-supervised model would be more data efficient (with respect to labelled data) than a supervised baseline.

**Data.** As mentioned, the domain in which we tested our methods is called the Playhouse, which originated in Interactive Agents Team [22]. The authors compiled approximately three years worth of interaction data, comprising about 2 billion image frames. The authors have confirmed no personally identifiable information or offensive content is contained in the dataset and gave consent for us to access it. As there are no associated captions with any frames, we treat each frame in this dataset as a single image, and used all the data as our "unpaired" dataset.

For the "paired" dataset, we engaged with crowd raters to provide corresponding captions for 78K uniformly sampled images from the unpaired dataset. Raters were instructed to describe what they saw in the image, with particular reference to a randomly selected object indicated in a bounding box. Detailed instructions are included in Appendix A.2. On average, each image contain 12.9 objects, and its corresponding caption describe 1.2 of them.

**Evaluation.** We measured the log probability of captions in the validation set, CIDEr score [30], object precision and "color-object pair precision". As the images were generated from the Playhouse environment, we had access to the ground truth object identities and colors present in each image, allowing us to check if any object and its color mentioned in a caption were indeed present in the image. We calculated object precision to be the proportion of correct objects among all mentioned in sampled captions, normalized by the same calculation from human captions. The color-object pair precision is calculated in the same manner, except for any mentions of an object and its color. As there are usually many objects in an image and captions tend to mention just a few, the recall counterpart of this metric (number of correct color-object pairs divided by those present in the image) is not very informative. Captions were greedily sampled for the calculation of CIDEr score object precision and color-object pair precision.

**Results.** Figure 3 shows sample data used to train the our models, as well as samples generated from the generative semi-supervised model. Models were able to both produce coherent and relevant captions given an image (e.g., "I can see a white bed where a green ball and a green duck are on it") and also generated realistic visual depictions of the Playhouse when conditioned on a language description.

Both semi-supervised methods performed better than the supervised baseline across log prob, CIDEr and color-object pair precision, including settings where we artificially constrained the size of the labeled dataset (Figure 4). Notably, when using the full amount of data our semi-supervised methods exceeded human color-object pair precision. To further put the performance gain in context, by leveraging the unpaired dataset of images the semi-supervised methods performed at a level equivalent to training a supervised model on $3\times$ more labeled data [31].

Interestingly, the semi-supervised methods perform similarly in the object precision metric, despite an advantage in color-object pair precision. This result suggest that the semi-supervised methods may have learned to bind color to objects from the unlabelled images, and this results in a performance gain compared to pure supervised methods.

### 3.2 Learning captions of new objects

**Hypothesis.** Upon validating our methodology and amassing the relevant data, we were well-poised to ask the following questions: do semi-supervised methods of intra-agent speech allow a model to quickly learn to speak about a new object with little supervision?

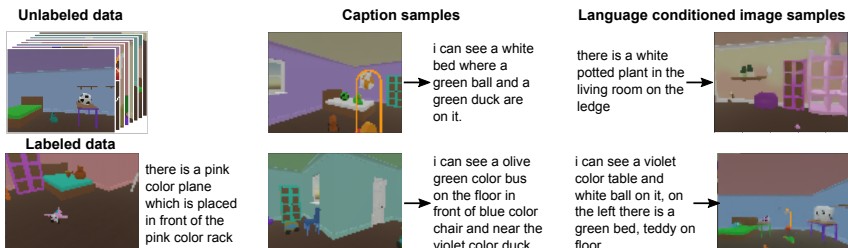

**Figure 3: Data and Model Samples.** Depicted are data used to train the semi-supervised model (left) and samples from the caption model (middle) and generative image model (right). As seen from the captions, generative modeling of images using a pre-trained language prior, in conjunction with auxiliary supervised training on a small number of labeled samples, is sufficient to produce meaningful and relevant language captions of images. Moreover, the model is able to generate realistic-looking scenes from provided captions. See Appendix A.3 for more samples.

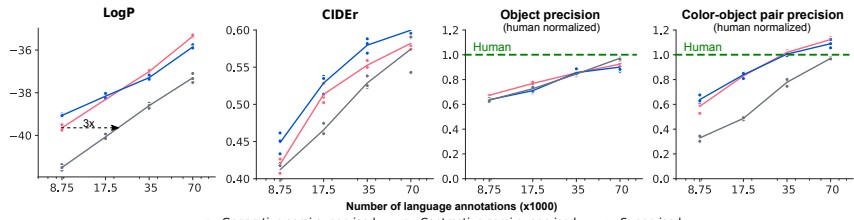

**Figure 4: Scaling performance of semi-supervised models over labeled dataset sizes.** Semi-supervised methods are trained with unlabeled data and titered amounts of labeled image-caption pairs. Here we calculate the log probability of validation set captions, the CIDEr score [30], object precision and color-object pair precision. Object precision (human normalized) is calculated by dividing the probability of any mentioned object in the image with that of the human ground truth. Color object pair precision (human normalized) is calculated similarly, except for any mentions of an object with its color. Lines represent means from 3 random seeds.

**Data.** To address the hypothesis, we first chose an object – a *drum* – and filtered our labeled caption data to remove all instances where the drum is either depicted visually or spoken about verbally. While not strictly necessary, from the unlabeled dataset we also removed all instances where the underlying (and unobserved) instruction included mention of a drum, which ensured that all instances of drum appearances were truly "passive". For example, the unlabeled data may have included trajectories of experience wherein a players was tasked with removing a ball from a shelf, but there might have also happened to be a drum in view in the room.

**Evaluation.** We evaluated the model using three different metrics. We first created a validation dataset of all captions mentioning drum with their associated image, and evaluated the model's average log likelihood on this drum dataset (drum caption log likelihood). The drum recall, which is the probability that them model mentions a drum given that a drum is present, measures how the model recognizes drums in the images. We also calculate its counterpart, the probability that a drum is present given a drum is mentioned, as the drum precision, to measure the accuracy of any drum mentions from the model. The perfect scores for drum recall and drum precision are 1 as a result of titering the drum data: we have removed all the data where drum is in the image or caption, and added back a small amount where drum is mentioned in the caption. This has a consequence that for any image with drum (although very few), its caption will always mention drum.

**Results.** We then performed the same experiment as in Section 3.1 at varying levels of re-introduced labeled drum data, as seen in Figure 4. As expected, the model's ability to speak about drums scales with the amount of labeled drum data with which it is trained. Notably, the model's recall reaches approximately 70% given only 585 labeled samples of drums. In contrast, the supervised method, where the new drum data is trained in concatenation with the non-drum data, reaches just over 20%. The increase of drum recall in the semi-supervised method is also accompanied with an increase of precision, where the semi-supervised method reaches 80%, versus 40% of the supervised method. These results indicate that the semi-supervised methods are able to learn about new objects, *implicitly*, by virtue of their language-conditioned reconstruction objective. That is, the mere presence of drums

in the visual observations was sufficient, in conjunction with a small amount of labeled drum captions, to elicit linguistic understanding of drums as relevant objects in the Playhouse.

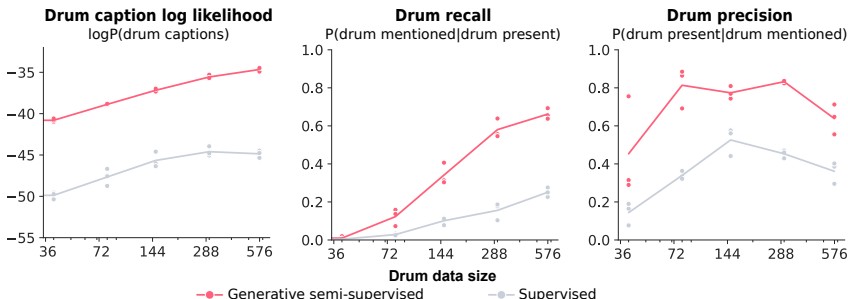

Figure 5: **Scaling performance of our semi-supervised methods over drum dataset sizes.** We examined how the semi-supervised and supervised methods scaled across different amounts of labeled data for a novel object. The semi-supervised trained model had a higher log likelihood, was more likely to recognize a drum in the image (high true positive rate) while displaying a comparable error-rate (same false positive rate). Extrapolating from supervised true positive rate, the supervised method would need more than $300\times$ more data to reach the same performance as the semi-supervised model. Lines represent means from 3 random seeds.

### 3.3 Learning to manipulate a new object

**Hypothesis.** As a final test we tackled the final question: Can the ability to quickly learn to speak about an object confer advantages for learning behaviors involving the new object (such as manipulating the object, or answering questions about it)?

**Methods and data.** To answer this question we leveraged the semi-supervised models built in Section 3.2 to provide auxiliary intra-agent speech objectives. While we employed a two-stage process (learning to caption, and subsequently learning from those captions to shape behavior) for expedience and ease of experimental setup, in principle the two learning procedures could occur in parallel in a unified agent. The agent then proceeded with behavioral cloning-based training as in Interactive Agents Team [22] on data that was filtered *to not include* any drum-based instructions (resulting in $9,344$ removed interactions). While after filter the dataset do not contain any demonstration of drum interaction, in some episodes the drum object is visible (larger than a pixel) in the background in 4.4% of the image frames.

**Evaluation.** We evaluated agent performance on two tasks: in the "lift drum" task, the agent receives the instruction "lift a drum", and is required to find the drum in a randomized Playhouse environment and lift it. In the "ask color drum" task, a single drum is spawned in the environment. The agent receives the instruction "what is the color of the drum?", and is then required to output language that answers the question (See Appendix A.4 for details). At the end of each episode, the task environment gives out a score of 0 or 1, indicating whether the task is completed successfully (note that these scores are used for evaluation purposes only, and not for training). We collected human performance on each of the tasks. Our participants receive an average of $0.98$ for the "lift a drum" task, and $0.51$ for the "ask color drum" task. The low score of "ask color drum" is primary due to the stringent reward function that require exact match of the canonical color name, and the fact that many colors are similar to the canonical name (for example, the canonical "cyan" color is can often referred as "light blue" by human participants). We calculate the final score by dividing the agent score by the average human score, so that a final score of 0 represents no success, and a 1 represents human-level performance.

**Results.**

As seen in Figure 6, a baseline agent trained without any drum-based instructions performs poorly on drum-based evaluation tasks, indicating that there is minimal transfer to be had when learning from the base interaction dataset. On the other hand, a model trained with drum-based instructions (that is, one that trains on the $9,344$ episodes that we had filtered) displays an expected upper bound for this agent of just over $75\%$ on "lifting drums", and just under $80\%$ for answering questions about a drum's color, similar to previously reported results [22].

Agents trained with both the caption loss and caption-matching loss, but *without any drum-based interaction data* achieve approximately $70\%$ of the performance attained by models trained directly on drum-based interaction data. To confirm the generalisability of our methods, we repeated our experiments on the teddy bear object, and observed similar result (Appendix A.5).

This is a substantial increase over the baseline model, and directly indicates that learning to speak about an object, which is afforded by training a model on as few as $585$ labeled examples using semi-supervised methods, allows an agent to subsequently exhibit behaviors that implicate the manipulation or mention of the object, without affecting performance on other known objects (Appendix A.5).

We then tested whether we could reduce the number of labeled captions even further, and observed that with as few as $150$ labeled captions we observed similar performance on drum-color question answering tasks, and only a slight decrease in drum lifting task performance.

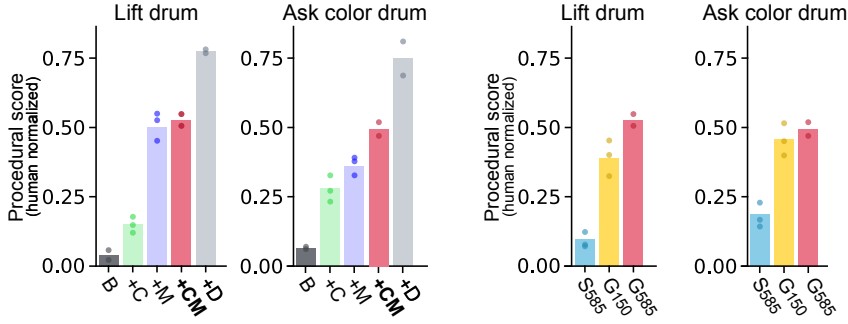

**B** = baseline without drum interaction, **+C** = caption loss, **+M** = caption matching loss,
**+D** = drum interaction data, **S$_{585}$** = supervised caption model with 585 annotations of drum,
**G$_{\{150,585\}}$** = generative semi-supervised caption model with {150, 585} annotations of drum

**Figure 6: Zero-shot task-directed interaction with a novel object.** We evaluated agent performance on two tasks that require interaction with the novel object (drum): in the first, the agent was tasked to locate and lift a drum, and in the second, the agent is required to mention the color of the drum using language. Our model with both the caption loss and caption matching loss (+CM) reaches more than $0.5$ human normalized reward, comparable to approximately $70\%$ of the performance of an agent trained with an additional $9344$ episodes of interaction data including drums (+D; left two panels). This result also suggests the crucial role of semi-supervised training in zero-shot generalization (right two panels): with just $150$ labeled captions of drum (G150), our model only displays a slight decrease in performance, but still significantly outperforms the supervised model trained on all the labelled captions (S585).

While we created two procedural tasks to quantitatively evaluate the agent, due to the diversity of the interaction data we used to train the agent, we hypothesized that the agent can perform more tasks with the novel object. Figure 7 summarizes a typical interactive session with our main agent ("+CM" from Figure 6), where the agent is asked to manipulate the novel object drum, and answer questions about it. The agent can recognize and manipulate the novel object under a variety of tasks. This further confirms that our method is general to many background tasks that the agent is able to perform.

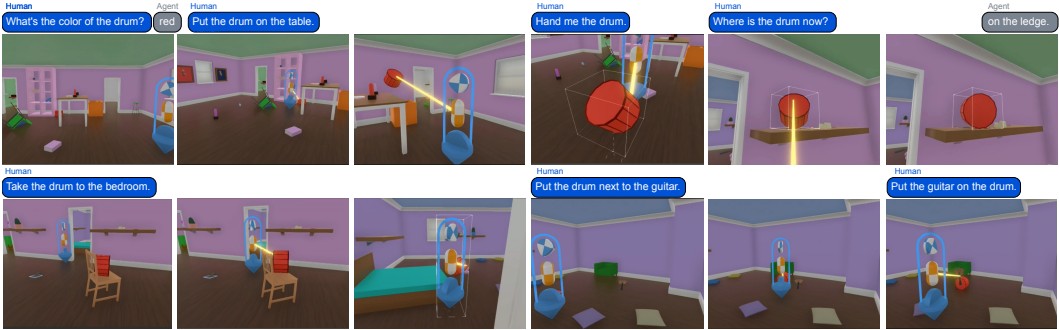

**Figure 7: Sample frames from human-agent interaction.** With our caption matching loss and caption loss, the agent trained without any drum-based instructions is able to manipulate the novel object drum and answer related questions under a variety of tasks.

# 4 Related Work

**Components.** The modeling in this study builds on components developed for generative modeling (VQVAE-2: [32]) and contrastive learning (CLIP: [20]).

**Task generalization in embodied agents.** Our work is in the tradition of grounded language research in virtual environments [33–35]. Recently, there have been efforts to demonstrate zero-shot task acquisition using a variety of means including with the use of pretrained models for language encoders [36] or visual-language models for zero-shot navigation [37]. The majority of these works have focused on generalization to novel phrasings of task instructions [38, 39] with some recent studies exploring generalization to new tasks given background knowledge from previous tasks [22, 40]. Similar to phenomena of zero-shot generalization in supervised classification and object detection [20, 21, 41], these studies have often relied on a large, diverse dataset providing supervision – where held-out tasks represent a novel recombination of parts of the training dataset (e.g., "compositional" generalization or systematicity [42]). Another line of research has focused on few-shot generalization [43–46] through meta-learning over a distribution of tasks – with generalization to other tasks within the distribution's support. Our work is complementary to both lines of work – and presents a means by which a different learning objective can embody "mental practice" enabling generalisation.

**Semi-supervised learning with natural language.** The semi-supervised methods we used to train the intra-agent speech module are related to many previous methods that use language as a latent variable. For example, Kingma et al. [13] developed a semi-supervised, variational autoencoder (VAE) with discrete latent variables, and this work has also been extended to use language as the latent variable. VAEs with language latent variables have also been studied in language summarization [47], and in the context of multi-agent communication with natural language [48, 49]. Our work may present a valuable contribution to this area, showcasing the utility of modern generative and contrastive models in supporting data-efficient learning. Our work's analysis of the scaling curve [31] properties of semi-supervised captioning [50–52] may provide additional analytical tools for the field.

**Language-conditioned image generation.** The use of language to create sophisticated, controllable generative models has seen rapid development in recent years. Recent works based on diffusion models [53–55] and autoregressive models have produced some of the most impressive results. Notable work includes the DALL·E family of models [56, 57] and GLIDE [58]. Other work in the area has also employed VAEs [59] and GANs [60–64].

# 5 Conclusions and Future Work

In this work we built a model for intra-agent speech by leveraging semi-supervised learning techniques. We have first shown that the semi-supervised methods outperforms the supervised model in captioning by leveraging unlabeled images. We further demonstrated that with 150 captions of a novel object, such models can help embodied agents execute interactive tasks with the novel object without having to learn from direct interactive experience (zero-shot), reaching 70% performance of a baseline agent trained with $9,000$ interaction episodes.

Although we developed the method in a virtual environment, it can be easily generalized to real world applications, applied with either good or bad intent. Embodied language agents acting as robots could be instructed to perform a wide manner of tasks, including harmful ones. Thus, if this work is to be applied in real settings, we advise extra caution to minimize unintended or harmful consequences.

This work is a first step into leveraging ideas from the psychology of inner and private speech in humans. While our source of speech was captioning, humans have a richer repertoire of techniques for intra-personal dialogue that affect modalities beyond vision. Direct next steps are, therefore, to explore a possible role of intra-agent speech in abstracting over time, and observing its effects on planning and reasoning.

## Acknowledgments and Disclosure of Funding

The authors would like to thank Felix Hill for critical discussion, Nathaniel Wong for help with the development of scripted probe tasks, Chris Dyer and Daan Wierstra for reviewing and comments on the manuscript, and Alex Goldin and Guy Scully for organizational support.

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
