# OpenReview forum: "Intra-agent speech permits zero-shot task acquisition"
_NeurIPS.cc/2022/Conference — NeurIPS 2022 Accept_

### Official Review · Reviewer_3uFs · 2022-07-08

**Rating:** 4
**Confidence:** 3
**Soundness:** 2 fair
**Presentation:** 3 good
**Contribution:** 3 good

**Summary:**

The authors present an "intra-agent speech" framework combining a semi-supervised image captioning model (herein M1) and an agent grounded in the Playhouse environment which takes action and answers questions (herein M2). M1 is trained first on a small amount of labeled data in the Playhouse domain, and its outputs on novel Playhouse samples are used to provide supervision to M2 -- an implementation of the "intra-agent speech" idea. The authors present results on 1) M1's semi-supervised captioning data efficiency, 2) M1's low-shot captioning data efficiency for novel objects, and 3) M2's low-shot performance for novel objects. They argue that this implementation of intra-agent speech provides data-efficient supervision to M2 for both language use and acting in the world.

**Questions:**

1. **Model design:** What is the motivation for your intra-agent speech implementation? A more obvious setup (to me) would be to have the outputs of a captioning model directly affect the representation of the agent, rather than serving as supervision in an auxiliary task. This would seem to resemble more closely the "intra-agent speech" idea -- in that the inner language is part of / directly influences mental representations online. While I'm not so attached to this alternate implementation, it is important for me to know why you chose your particular implementation. Can you explain why you chose this auxiliary task route? Why might it be the most effective and/or cognitively faithful method for implementing intra-agent speech?
	1. For what it's worth, I'd strongly prefer a thorough discussion on this design choice and alternatives in place of the ~1.5 pages dedicated to the generative vs. contrastive training setup.
2. **3.1 evaluation:** Why is "color-object accuracy" the right custom metric to develop here? I have many other more stringent questions I'd want to ask about a captioning model in this domain. For example:
	1. Of the concrete nouns mentioned in a caption, how many have referents in the scene?
	2. For noun phrases "the X," how often is there a unique referent selected by X in the scene?
3. **3.2 training:** What is the training method here? Is the model being fine-tuned with drum examples, or retrained afresh with a dataset that is the union of the original dataset with some fixed number of drum examples?
	1. Possibly related question -- what happens to the log-probability of captions for non-drum images as the drum-training proceeds?
4. **3.2 evaluation:**
	1. What is the ideal "drum true positive rate?" Captions needn't include every object in a scene. / Humans won't always mention a drum when it appears in an image. Why is 70% better than 20%?
	2. The drum false positive rate result doesn't look so interesting to me. There may not be any significant differences between semi-supervised and supervised training. I'd rather see here the other conditional P(drum present | drum mentioned).
5. **3.3 evaluation:**
	1. Over what trajectories are human reward labels collected? Does it include ground-truth captions and ground-truth motions? This seems very important for making sense of the absolute value of the rewards shown in figure 6.
	2. Can you show the full distribution of human reward labels, without normalization? (At least in an appendix.)
	3. How does the "ask color" task work? This doesn't seem analogous to the captioning task imposed on the agent during training.
6. **Arbitrary evaluation choices:**
	1. Why drums? Do you plan to repeat the analyses for other objects? You could have a leave-one-out style evaluation for all objects that appear (or appear with low frequency, at least) in the unlabeled data.
	2. Why just lifting? How many motion tasks are there available to test?
		1. What about a task that is more semantically complicated (e.g. "put the drum on the table") or motorically complicated (e.g. "balance the drum on its side")?
7. **Interpretation:** Why does this work? I think the paper would be much more compelling if the authors could provide detail (through a model comparison, representational analyses, or error analysis, for example) on what the auxiliary captioning objective (AKA intra-agent speech) is doing for the model. The particularly interesting result is that the captioning objective affects not just the linguistic output of the agent but also its motion behavior.

**Limitations:**

Yes.

**Strengths And Weaknesses:**

Overall, the paper presents an interesting idea motivated by a psychological concept, but the execution and presentation are far from thorough. The model design and evaluation seem to contain several arbitrary choices, and little analysis in the paper helps the reader understand *why* the model improves concretely.

I do not follow this subfield closely and can't speak to the originality or significance of this paper with confidence. I do regard it as significant in that it makes a good-faith attempt to implement a psychologically motivated concept.

---

> ### Author Response · Authors · 2022-08-02
> **Thanks for reviewing!**
>
> >The model design and evaluation seem to contain several arbitrary choices, and little analysis in the paper helps the reader understand why the model improves concretely.
> >Model design: What is the motivation for your intra-agent speech implementation? A more obvious setup (to me) would be to have the outputs of a captioning model directly affect the representation of the agent, rather than serving as supervision in an auxiliary task. This would seem to resemble more closely the "intra-agent speech" idea -- in that the inner language is part of / directly influences mental representations online. While I'm not so attached to this alternate implementation, it is important for me to know why you chose your particular implementation. Can you explain why you chose this auxiliary task route? Why might it be the most effective and/or cognitively faithful method for implementing intra-agent speech?
> >For what it's worth, I'd strongly prefer a thorough discussion on this design choice and alternatives in place of the ~1.5 pages dedicated to the generative vs. contrastive training setup.
>
> There are multiple ways one can use information to influence an agent's representations, including using an auxiliary loss (i.e., shaping through gradients) and using a conditioning vector (i.e. shaping by changing the functional form of the network). While both approaches are equally reasonable, we chose the implementation purely from a practical perspective: anecdotally we have found it more effective to use an auxiliary loss in other similar behavior cloning experiments, where conditional inputs can often get ignored by agents.
> In addition, using an auxiliary loss has the practical benefit that allows us to use a larger and better caption model to produce intra-agent speech samples. The auxiliary loss implementation not only makes it easy to train inner speech at a slower rate (in our implementation it is only applied once every 25 steps) without slowing down agent behavioral training, but also no longer required for inference and simplifies our evaluation pipeline.
>
> >3.1 evaluation: Why is "color-object accuracy" the right custom metric to develop here? I have many other more stringent questions I'd want to ask about a captioning model in this domain. For example:
> >Of the concrete nouns mentioned in a caption, how many have referents in the scene?
> >For noun phrases "the X," how often is there a unique referent selected by X in the scene?
>
> We would like to thank the reviewer for these suggestions. We are now reporting the reviewer’s suggestion 1 as “object precision” in the results of Section 3.1. We also changed the name of “color-object accuracy” to “color-object pair precision” for clarity.
>
> The color-object pair precision is a more difficult metric, as it not only requires the model to correctly identify the object and the color, but more importantly, also needs the model to understand that a specific color is bound to a particular object. In fact, the semi-supervised methods perform similarly to the purely supervised methods in object precision, however they show an advantage in the color-object pair precision.
>
> This suggests that unlabelled images helped the semi-supervised model to learn object-color binding, which the supervised model struggles with limited amounts of data. We have also revised the results of Section 3.1 to include this discussion.
>
> We did not implement the second suggestion from the reviewer (whether “the X” in the caption corresponds to a unique referent in the image). This is mostly because our main goal of using the caption model as inner speech is to teach the agent about the new object, and not about the concept of uniqueness. However, we believe the reviewer's suggestion is very interesting, and we can even generalize uniqueness to the concepts of counts: if we set out to collect a caption dataset of objects and their counts and use that as inner speech, would it be helpful to teach our agents about numbers? This would be a curious hypothesis to investigate.
>
> >3.2 training: What is the training method here? Is the model being fine-tuned with drum examples, or retrained afresh with a dataset that is the union of the original dataset with some fixed number of drum examples?
> >Possibly related question -- what happens to the log-probability of captions for non-drum images as the drum-training proceeds?
>
> The model is retrained afresh with both the original dataset and a fixed number of drum examples, and we do not see a significant difference in the log probabilities of non-drum captions in this training paradigm. We now clarified the training paradigm in Section 3.2.

---

> > ### Author Response · Authors · 2022-08-02
> > **cont'd**
> >
> >
> > >3.2 evaluation:
> > >What is the ideal "drum true positive rate?" Captions needn't include every object in a scene. / Humans won't always mention a drum when it appears in an image. Why is 70% better than 20%?
> >
> > This is an important piece of information that we missed in the manuscript. The ideal “drum true positive rate” (now renamed to “drum recall”) is 1 as a result of titering the drum data: we have removed all the data where drum is in the image or caption, and added back a small amount where drum is in the caption. This has a consequence that for any image with drum (although very few), the caption will always mention drum.
> > We appreciate the reviewer for pointing this out, and have revised evaluation of Section 3.2 to include this information in the manuscript.
> >
> > >The drum false positive rate result doesn't look so interesting to me. There may not be any significant differences between semi-supervised and supervised training. I'd rather see here the other conditional P(drum present | drum mentioned).
> >
> > We thank the reviewer for the suggestion and have now revised the manuscript to report P(drum present | drum mentioned) as “drum precision”. We have also renamed “drum true positive rate” as “drum recall” for more clarity.
> >
> > >3.3 evaluation:
> > >Over what trajectories are human reward labels collected? Does it include ground-truth captions and ground-truth motions? This seems very important for making sense of the absolute value of the rewards shown in figure 6.
> >
> > The evaluation is performed on a hand-crafted procedural task where the agent acts on itself, and receives a procedural score of 1 at the end of the episode if it succeeds, or 0 if it fails. Our agent is not trained on this evaluation task, and human scores are collected by letting humans play the task.
> > We have added explanations in Section 3.3 evaluation to clarify that in these evaluations the agents receive procedural scores given by the task environment.
> >
> > >Can you show the full distribution of human reward labels, without normalization? (At least in an appendix.)
> >
> > We now report the human score in “Methods and data”, section 3.3 and clarified that the final score is calculated by the raw agent score divided by the human score:
> > Our participants receive an average of 0.98 for the "lift a drum'' task, and 0.51 for the "ask color drum'' task. The low score of "ask color drum'' is primary due to the stringent reward function that require exact match of the canonical color name, and the fact that many colors are similar to the canonical name (for example, the canonical "cyan'' color is can often referred as "light blue'' by human participants). We calculate the final score by dividing the agent score by the average human score, so that a final score of 0 represents no success, and a 1 represents human-level performance.
> >
> > >How does the "ask color" task work? This doesn't seem analogous to the captioning task imposed on the agent during training.
> >
> > The “ask color” task is another hand-crafted procedural level that we used only for evaluation. In this task, the agent is spawned in the playhouse environment. The agent is then prompted with the instruction “What is the color of the [object]”? And for testing the agent’s performance, we used drums as the target objects.
> >
> > The reviewer is correct that the task is different from the captioning task. In this task we aim to test if the agent can utilize the new information learned from the intra-agent speech module (the drum), and generalize it into a separate language output task. And indeed, we found the intra-agent speech allows the agent to zero-shot complete this task with the new object.
> >
> > >Arbitrary evaluation choices:
> > >Why drums? Do you plan to repeat the analyses for other objects? You could have a leave-one-out style evaluation for all objects that appear (or appear with low frequency, at least) in the unlabeled data.
> >
> > The drum is a randomly chosen object in the environment. We are currently working on repeating our approach on another novel object, as well as on a novel color. The training process unfortunately will take several weeks, we will report intermediate results here, and include the final result in the camera ready version.
> >
> >
> > >Why just lifting? How many motion tasks are there available to test?
> > >What about a task that is more semantically complicated (e.g. "put the drum on the table") or motorically complicated (e.g. "balance the drum on its side")?
> >
> > It is hard to hand craft a procedural evaluation level for these complicated tasks, however we will include videos of the agent to demonstrate the agent’s ability in manipulating the novel objects.

---

> > > ### Author Response · Authors · 2022-08-02
> > > **cont'd**
> > >
> > >
> > > >Interpretation: Why does this work? I think the paper would be much more compelling if the authors could provide detail (through a model comparison, representational analyses, or error analysis, for example) on what the auxiliary captioning objective (AKA intra-agent speech) is doing for the model. The particularly interesting result is that the captioning objective affects not just the linguistic output of the agent but also its motion behavior.
> > >
> > > Our agent is trained with a background demonstration of interaction with known objects. For interacting with the novel object, the agent will need to form a language representation of the novel object’s name, bind the name of the novel object to its corresponding visual representation, and then transfer the known tasks to the novel object.
> > >
> > > We have applied two auxiliary losses to shape the language representation of the novel object both in language input and output, and bind to the visual input. We have shown that with a diverse background interaction with known objects, this is enough for the agent to apply known tasks to the novel objects.
> > >
> > > To test this hypothesis that our auxiliary losses shape the agent’s representation of the drum, we will train a decoder to decode all the objects in the image from the encoder representation of each agent. We hypothesize that the baseline agent that trained only on background data without interacting with the novel object will not have a good representation of it, and our auxiliary loss has shaped the representation of the novel object for the memory and policy.
> > >
> > > We will update when the results of these experiments come out.

---

### Official Review · Reviewer_2NEi · 2022-07-09

**Rating:** 5
**Confidence:** 2
**Soundness:** 2 fair
**Presentation:** 3 good
**Contribution:** 2 fair

**Summary:**

This paper is a novel proof-of-concept of embodied agents gaining the ability of new tasks with two kinds of semi-supervised learning mechanisms, namely 'intra-agent speech'. The results are three-fold - embodied interaction of agents within a simulated environment (or intra-agent speech of unified agent): (1) enhances the data efficiency of the image captioning subsystem of the agent, (2) reduces the number of data required to learn about the new object, (3) allows motor task involving the new object.

**Questions:**

- Where can I find the image captioning performance of the model trained with only semi-supervised pretraining (i.e., intra-agent speech without an agent)?
- What does the 'indicator embedding' in line 151 refer to?

**Limitations:**

The reviewer is unsure whether this work shows that the inner speech of the agents is helpful in efficiently learning about the world as humans do. Unless there is a sufficient amount of experiments and literature in a similar field, such over-claiming should be avoided.

**Strengths And Weaknesses:**

Strengths
- This work largely unifies the frameworks from image captioning, image generation, and reinforcement learning to show the newly devised concept 'intra-agent speech' can work as an effective semi-supervised learning mechanism.
- The writing is clear enough for readers to understand what hypotheses are established and how they are tested.

Weaknesses
- The conclusions derived from the experiments aren't backed with enough numerical details. For instance, how should the readers conclude that the proposed methods allow a model to 'quickly' learn with 'little' supervision? To meet a broader audience, those ambiguities should be resolved.
- The zero-shot skill acquisition task (section 3.3) is only experimented with a single specific object (drum), which is not convincing enough.
- 95% confidence intervals are obtained with only 3 experiments, which does not seem to be enough sample size. The reviewer considers 5 as a bare minimum for such statistics. (It's also mistyped - "confident interval" should be "confidence interval")

---

> ### Author Response · Authors · 2022-08-02
> **Thanks for reviewing!**
>
> >The conclusions derived from the experiments aren't backed with enough numerical details. For instance, how should the readers conclude that the proposed methods allow a model to 'quickly' learn with 'little' supervision? To meet a broader audience, those ambiguities should be resolved.
>
> We have now revised the manuscript to remove any ambiguities in our claims and replace them with quantitative statements. I.e. in the conclusion section: “In this work we built a model for intra-agent speech by leveraging semi-supervised learning techniques. We have first shown that the semi-supervised methods outperforms the supervised model in captioning by leveraging unlabeled images. We further demonstrated that with 150 captions of a novel object, such models can help embodied agents execute interactive tasks with the novel object without having to learn from direct interactive experience (zero-shot), reaching 70% performance of a baseline agent trained with 9,000 interaction episodes.”
>
> >The zero-shot skill acquisition task (section 3.3) is only experimented with a single specific object (drum), which is not convincing enough.
>
> We are setting up a second zero-shot transfer experiment with another object and testing if the method generalizes across unseen color categories as well object categories. These experiments will unfortunately take several weeks to complete, so we will include them in the camera-ready version if the paper is accepted.
>
> >95% confidence intervals are obtained with only 3 experiments, which does not seem to be enough sample size. The reviewer considers 5 as a bare minimum for such statistics. (It's also mistyped - "confident interval" should be "confidence interval")
>
> We agree with the reviewer that a sample size 3 is quite small, however experiments with agents at this scale already take significant computation resources. We have now updated Figures 4, 5 and 6 to remove the 95% confidence intervals and plot individual data points instead.
>
>
> >Where can I find the image captioning performance of the model trained with only semi-supervised pretraining (i.e., intra-agent speech without an agent)?
>
> The semi-supervised pretraining performance are summarized in Figure 4 for overall performance, and Figure 5 for learning a novel object.
>
> >What does the 'indicator embedding' in line 151 refer to?
>
> We have now clarified this in the revised manuscript, line 153-154 “the input of the language policy is summed with a learnable embedding of an indicator variable representing whether the target comes from the pre-trained caption model or from the language output in the human demonstration data.”
>
> >The reviewer is unsure whether this work shows that the inner speech of the agents is helpful in efficiently learning about the world as humans do. Unless there is a sufficient amount of experiments and literature in a similar field, such over-claiming should be avoided.
>
> This is an interesting point, but we believe in the manuscript we do not claim that our efficiency is comparable to humans, or whether inner speech in humans contributes to data efficient learning. However we do aim to convince the reader that if we build a model that reflects some properties of inner speech, we can achieve greater data efficiency. In this sense we can empirically provide a computational justification or rationale for inner speech.
> However, if there are specific parts of the manuscript that are misleading, we would be more than happy to revise.

---

### Official Review · Reviewer_G4n2 · 2022-07-11

**Rating:** 8
**Confidence:** 4
**Soundness:** 4 excellent
**Presentation:** 2 fair
**Contribution:** 3 good

**Summary:**

In this paper, the authors model intra-agent speech as latent codes for embodied agents. In optimizing the ELBO, the authors use a frozen language model to enforce a nautal language like latent code distribution. They test this model on image caption tasks with both paired data and unpaired data, and find with the same amount of unpaired data, semi-supervised could benefit from unpaired data. They also find that the model can learn to speak about objects that are rare in the paired data and only passively observed in the unpaired data. Finally, they find that just through adding “drum” related captioning data, the agent can learn to lift drum.


**Questions:**

Most of the zero-shot results are only on a single concept -- drum. What are the obstacles preventing the authors from conducting the experiments on other concepts as well?

**Limitations:**

Yes

**Strengths And Weaknesses:**

Strengths

The intra-agent speech concept has rarely been discussed in the embodied agent literature.
Their finding about generalizing to zero-shot interaction tasks from captioning data is very interesting.

Weakness:

Using variational methods for semi-supervised learning is not a new idea. The first finding on the benefit of unpaired data is expected. This may not necessarily be a weakness, but branding it as “intra-agent speech” might be misleading and hyping. Since there’s no evidence that this is similar to the human inner speech, it should be made clear that this work has nothing to do with inner speech, and is just a latent variable model. My scores are based on the assumption that the authors could make the amendment to make this clear.

---

> ### Author Response · Authors · 2022-08-02
> **Thanks for reviewing!**
>
> >Using variational methods for semi-supervised learning is not a new idea. The first finding on the benefit of unpaired data is expected. This may not necessarily be a weakness, but branding it as “intra-agent speech” might be misleading and hyping. Since there’s no evidence that this is similar to the human inner speech, it should be made clear that this work has nothing to do with inner speech, and is just a latent variable model. My scores are based on the assumption that the authors could make the amendment to make this clear.
>
> We agree with the reviewer that using variational methods in semi-supervised learning is not a new idea. However, we hope that our particular method using combinations of language priors, and generative and contrastive decoders was, however, a technically interesting and successful implementation of the idea.
>
> We also agree with the reviewer and do not view the semi-supervised learning method alone to be the foundation for connecting the method to “inner speech”. The second part of the work dealt with using an inner commentation process – captioning of experiences – to improve agent performance. We hope this secondary process of inner rehearsing, depending on our semi-supervised captioning model, makes the model interesting from the perspective of cognitive science and a useful model of “inner speech”.
>
> We have revised Section 2 to include this explanation.
>
>
> >Most of the zero-shot results are only on a single concept -- drum. What are the obstacles preventing the authors from conducting the experiments on other concepts as well?
>
> This is a great question. We are setting up a second zero-shot transfer experiment with another object and testing if the method generalizes across unseen color categories as well object categories. These experiments will take several weeks to complete, so we will include them in the camera-ready version if the paper is accepted.

---

### Official Review · Reviewer_Dfhj · 2022-07-12

**Rating:** 5
**Confidence:** 4
**Soundness:** 3 good
**Presentation:** 3 good
**Contribution:** 3 good

**Summary:**

This paper proposes using image captioning (i.e. intra-agent speech) as an auxiliary task to help embodied agent to execute tasks involving novel objects which are not in the training interaction dataset. In order to provide captions as the target for training agent’s policy, this paper proposes two semi-supervised methods to learn captioning models. The experiments show that 1) the semi-supervised methods can learn to caption with fewer data compared to supervised learning and 2) learn to caption a new object from hundreds of examples to shape the policy with the learned captions to interact with that novel object.

**Questions:**

- In the experiment of section 3.3, the authors only use the drum data to train the caption model and then use this caption model to train the agent without drum interaction data. In the second training stage, does the agent see and caption the drum? If not, it is unlikely the agent can learn to interact with this new object. How much interaction data in this experiment contains a drum as a background object? This would be a key contributor to the zero-shot performance.
- Do the captions cover most of the visible objects in the scene? The proposed approach will perform better if the caption has higher coverage of objects. How robust is the proposed model when having lower object coverage?

**Limitations:**

- As the authors suggest, the proposed intra-agent speech can only generalize to object identification tasks as the captions and observations are visual and object centric.


**Strengths And Weaknesses:**

Strengths
- Novelty: The idea to use captioning as a representation to improve policy generalization is interesting.
- Data efficiency: The proposed semi-supervised method can caption images with better performance with fewer annotations compared to supervised learning.
- Explainability: The auxiliary captioning task can potentially help explain the agent’s perception and relates the linguistic representation with the generated policy. However, the paper doesn’t show how the generated captions relate to the success or failure of the agent’s policy.

Weaknesses
- For the proposed agent to work, it requires specific setups which may be easy to break.
  1) the caption model needs to be relevant to the tasks the agent will perform. In this paper, they are about identifying objects and their colors. So, the object and color information must be included in the caption annotations the example annotation shows. It would be helpful to see how the agent’s performance relates to how captions match the agent’s task.
  2) the target object needs to be in the background (see Questions), it is unclear how many times the agent needs to see the object in the background in order to interact with it

---

> ### Author Response · Authors · 2022-08-02
> **Thanks for reviewing!**
>
> >the caption model needs to be relevant to the tasks the agent will perform. In this paper, they are about identifying objects and their colors. So, the object and color information must be included in the caption annotations the example annotation shows. It would be helpful to see how the agent’s performance relates to how captions match the agent’s task.``
>
> The goal of this paper was to establish a method that allows knowledge transfer between  language descriptions to embodied behavior. Indeed, information about the task must be present in the captions; otherwise, there is little that we can hope that language descriptions will contribute to embodied behavior. This is hinted at from the performance of the supervised model: even if the model can caption all other objects very well, it still does not contribute to performance on drum tasks.
>
> We don’t regard this as a drawback: note that learning from a caption of the scene is a step toward the human-like developmental learning challenge of learning from a parent or tutor who speaks descriptively in a shared environment. Thus, this work is a step toward more “third-person” learning instead of learning entirely from first-person demonstration data.
>
> However, we note that the captions themselves have little to do with the motor requirements for the tasks that we demonstrate transfer to (such as  lifting, handing, etc.) , since these verbs are not mentioned in the captions.
>
> To demonstrate that the agent can apply a diverse set of known skills to a new object, we will also include videos of the agent zero-shot performing more complicated tasks with a new object in the supplemental material.
>
> >the target object needs to be in the background (see Questions), it is unclear how many times the agent needs to see the object in the >background in order to interact with it
> In the experiment of section 3.3, the authors only use the drum data to train the caption model and then use this caption model to train the agent without drum interaction data. In the second training stage, does the agent see and caption the drum? If not, it is unlikely the agent can learn to interact with this new object. How much interaction data in this experiment contains a drum as a background object? This would be a key contributor to the zero-shot performance.
>
> We thank the reviewer for raising this interesting question. In the second stage of training, the agent can see background drum objects and can therefore produce captions referring to them. This “intra-agent speech” produces learning targets that shape the agent’s representations. We agree with the reviewer’s comment that this requires background encounters with the novel object. The drum is present in 4.4% of the video frames in the interaction data and we have revised “Methods of Data” in Section 3.3 to include this information for the readers.
>
> However, in this work, we investigated the semi-supervised regime, where the background unlabelled data is inexpensive and abundant and the language labels are scarce and limited. We find this setup interesting as it is what humans face during development, where linguistic input is sparse compared to non-linguistic sensory data (a kind of “poverty of the stimulus”). We agree that more careful consideration of the scaling performance as a function of background data would also be interesting, but we do feel that it is the less important analysis for our interests.

---

> > ### Author Response · Authors · 2022-08-02
> > **continued**
> >
> >
> > >Do the captions cover most of the visible objects in the scene? The proposed approach will perform better if the caption has higher coverage of objects. How robust is the proposed model when having lower object coverage?
> >
> > We thank the reviewer for pointing this out. The captions in our dataset already describe a small number of objects per image. On average, each caption on average mentions 1.2 objects out of 12.9 objects for each image. Moreover, since we collected the caption dataset by prompting the participants to describe a random object in view, the caption can often focus on objects far away, and this can represent an even smaller proportion of pixels in the image. Therefore we believe our model is robust against low object coverage.
> >
> > We have revised the data section of 3.1 to include the coverage information.
> >
> > >However, the paper doesn’t show how the generated captions relate to the success or failure of the agent’s policy.
> >
> > In our approach we treat the intra-agent speech as an auxiliary loss, therefore during inference, no intra-agent speech is sampled, and the policy is not directly conditioned on the generated captions.
> >
> > While the generated captions do not directly affect the agent’s policy at each timestep, the quality of the captions during training as a whole affects the agent’s generalization performance. As shown in Figure 6, agents with better caption models have a higher zero-shot performance on our evaluation tasks, suggesting that the quality of the caption model is critical for zero-shot generalization.

---

### Author Response · Authors · 2022-08-02
**General response**

We thank the reviewers for their helpful comments and feedback, and we largely agree with all of the suggestions. We are quite excited that many reviewers have found the line of research of modeling inner speech in embodied agents to be interesting and novel. The reviewers have additionally raised some common questions and comments that we paraphrase below:

- How does the algorithmic method link to the concept of inner speech in psychology?
- How general is the method? Would it work for other tasks, objects, or concepts?
- The reviewers asked for some more detailed analyses and evaluations.

In response to these suggestions, we have made the following amendments to the manuscript:

- We have revised the approach sections to clarify how our model is inspired by inner speech.

- We have created videos showing several other kinds of interactions of the agent in more complicated scenarios. We are setting up a second zero-shot transfer experiment with another object and testing if the method generalizes across unseen color categories as well object categories. These experiments will take several weeks to complete, so we will include them in the camera-ready version if the paper is accepted.

- We have revised the evaluation metrics and provided rationales for these particular metrics. We are performing a decoder-based analysis to probe which object categories are recognized by the merged language and vision encoder. Finally, we will perform an experiment comparing imitation learning data efficiency against the semi-supervised intra-agent speech data efficiency, and we will update when the results come out.

We feel that the method is innovative and is of general interest to AI community members and cognitive scientists. We hope these revisions improve the manuscript and address the reviewers’ concerns.

---

> ### Author Response · Authors · 2022-08-03
> **Videos for agents**
>
> Please find the videos showing the agents respond to more complicated instructions about drum:
>
> intra agent speech:
> - success: https://drive.google.com/file/d/1EAKW4-BoZornGXKADU9o5kl0BNfC8tV5/view?usp=sharing
>   - "put drum on the table", "what's the color of drum"
> - success: https://drive.google.com/file/d/180aR6Y9rtDlVPv_WckfJP4yZZNDHJhfN/view?usp=sharing
>   - "take the drum to the bedroom", "put guitar on the drum"
> - mixed: https://drive.google.com/file/d/1cnjd6VvJZoAxvGFtiO_AG7R99Kt7iQGr/view?usp=sharing
>   - "hand me the drum"
> - fail: https://drive.google.com/file/d/1qDPp26BQQLl1y0ifflWaBL5eSeva-lcg/view?usp=sharing
>   - "put the drum on the table", "hand me the drum"
>
> Baseline (as expected, we are unable to get any success videos):
> - fail: https://drive.google.com/file/d/1UbDsTsc1kYDCl8JcvaUZwV60pfC_fvmB/view?usp=sharing
>   - "put green drum on the table", "lift the drum"
> - fail: https://drive.google.com/file/d/1lrYqeDQsCz4LkkL5itzDrW7f9sFCiS73/view?usp=sharing
>   - "what's the color of the drum", "where is the drum"

---

### Meta-Review · Area_Chair_tRAQ · 2022-08-31

**Recommendation:** Accept
**Confidence:** Certain

**Metareview:**

This work aims to provide a faithful translation of "inner speech" to a virtual agent via image captioning. This is integrated via an auxiliary loss and shown to assist in zero-shot transfer to a new object.  This aids in reasoning about the otherwise overly rich and potentially spurious visual space.

The good faith approach and efficient learning results were both viewed as strong positives.  The primary concerns revolve around the very limited number of samples and whether the approach can actually generalize beyond the limited evaluation domain presented here.  The integration of additional experiments promised will dramatically strengthen the work

**Award:**

No

---

### Decision · Program_Chairs · 2022-09-14

Accept